# Recent Advances in Metal–Organic Frameworks for the Surface Modification of the Zinc Metal Anode: A Review

**Yibo Xing [1], Kaijia Feng [1], Chunyang Kong [1], Guangbin Wang [1], Yifei Pei [1], Qixiang Huang [1] and Yong Liu [1,2,3,\*]**

[1] Henan Key Laboratory of Non-Ferrous Materials Science & Processing Technology, School of Materials Science and Engineering, Henan University of Science and Technology, Luoyang 471023, China; y2824372495@163.com (Y.X.); fengkaijia2021@163.com (K.F.); kcy00312@163.com (C.K.); guangbin2022@163.com (G.W.); 15738595833@163.com (Y.P.); hqx19990606@163.com (Q.H.)

[2] Longmen Laboratory, Luoyang 471023, China

[3] Provincial and Ministerial Co-Construction of Collaborative Innovation Center for Non-Ferrous Metal New Materials and Advanced Processing Technology, Henan University of Science and Technology, Luoyang 471023, China

\* Correspondence: liuyong209@haust.edu.cn

**Abstract:** Aqueous zinc ion batteries (AZIBs) are considered as one of the most promising energy storage technologies due to their advantages of being low in cost, high in safety, and their environmental friendliness. However, dendrite growth and parasitic side reactions on the zinc metal anode during cycling lead to a low coulombic efficiency and an unsatisfactory lifespan, which seriously hinders the further development of AZIBs. In this regard, metal–organic frameworks (MOFs) are deemed as suitable surface modification materials for the Zn anode to deal with the abovementioned problems because of their characteristics of a large specific surface area, high porosity, and excellent tunability. Considering the rapidly growing research enthusiasm for this topic in recent years, herein, we summarize the recent advances in the design, fabrication, and application of MOFs and their derivatives in the surface modification of the zinc metal anode. The relationships between nano/microstructures, synthetic methods of MOF-based materials, and the enhanced electrochemical performance of the zinc metal anode via MOF surface modification are systematically summarized and discussed. Finally, the existing problems and future development of this area are proposed.

**Keywords:** aqueous zinc ion batteries; metal–organic frameworks; surface modification of Zn anode; metal nodes of MOFs; electrochemical performance

## 1. Introduction

In the past decades, along with the rapid development of economy and society, nonrenewable energy sources have been consumed in large quantities and gradually exhausted, and the overuse of these nonrenewable energy sources has caused a series of environmental problems [1–4]. In this context, with the strengthening of environmental protection awareness and the emphasis on sustainable development, "carbon peak" and "carbon neutrality" have gradually become a common goal of human beings. Therefore, the exploration of new energy technologies, such as solar energy and wind energy, has become a hot research topic in recent years [5–7]. However, the intermittency and fluctuation of these energies greatly impede their further application [8–10]. In this regard, as an efficient electrochemical energy storage (EES) system, secondary batteries can be used as the intermedium for peak-load shifting to realize a more stable and reliable power supply [11–15]. Among them, lithium-ion batteries (LIBs) have received considerable attention and dominated the current EES market share because of their low self-discharge rate and long cycle lifespan [16–20]. However, some shortcomings of LIBs critically hinder their further development, especially for large-scale energy storage, such as the shortage of lithium resources and the use of flammable organic electrolytes, which can make it dangerous in the process of

usage [21–24]. Hence, it is urgent to develop low-cost and high-safety rechargeable battery systems as alternatives for LIBs.

In recent years, as a type of energy-storage device, aqueous zinc-ion batteries (AZIBs) have attracted wide attention from researchers due to their advantages, including being low in cost, high in safety, and environmentally benign [25–27]. And the zinc metal anode possesses several merits, such as a high earth abundance, high theoretical capacity (820 mAh g$^{-1}$ and 5855 mAh cm$^{-3}$), and a low redox potential ($-0.76$ V vs. SHE). However, the dendrite growth and interfacial side reactions of the Zn anode during cycling give rise to the unsatisfactory electrochemical performance of AZIBs in practical use [28–30]. During the cycles of AZIBs, an uneven electric field distribution and a Zn ion flux will lead to the formation and growth of Zn dendrites, which would ultimately penetrate the separator and short-circuit the batteries [7,31]. Moreover, corrosion and hydrogen evolution reactions (HERs) on the Zn anode further reduce the coulombic efficiency and lifespan of AZIBs [32,33]. In order to solve these problems of the Zn anode, researchers have made a lot of effort and put forward various improvement strategies; for example, electrolyte optimization [34], surface modification [35–37], 3D host structure design [38,39], and so on. Therein, surface modification, as one of the effective methods, can provide abundant nucleation sites, uniformize the Zn ion flux, and accelerate the desolvation process, which helps to inhibit dendrite growth and side reactions on the Zn anode to achieve high-performance AZIBs [40–42].

Metal–organic frameworks (MOFs) are a kind of crystalline porous material based on the coordination bond between metal ion/cluster nodes and organic ligands [43]. MOFs with a different structure and composition can be fabricated by combining different organic and inorganic parts or adopting different reaction conditions [44]. MOFs possess a large specific surface area, high porosity, and an adjustable composition and structure [45]. Additionally, the MOF derivatives obtained by the calcination with MOFs as a precursor not only retain the morphology and pore structure of MOFs [46], but also obtain other unique properties, which further broadens the application of MOFs. Benefiting from the above advantages, MOFs and their derivatives are widely used for electrode materials and surface-modification materials in energy-storage devices (e.g., lithium-ion batteries [47], sodium-ion batteries [48], and supercapacitors [49]). To date, MOF-based materials have received considerable attention for the surface modification of the Zn metal anode [50–53]. For example, in 2020, Yang et al. coated activated ZIF-7 on a Zn surface, which could promote the partial desolvation process before the reduction of Zn$^{2+}$ and thus greatly improve the electrochemical performance of AZIBs [50]. Later, by means of coating the UIO-66-SO$_3$H, Wang and coworkers successfully fabricated an artificial protective layer on a Zn anode, with which the AZIB batteries exhibited an enhanced cycling stability [52]. In addition, some reviews summarize the progress of MOFs in various aqueous energy device applications [1,54–56]. For instance, in 2021, Tan et al. systematically summarized the applications of MOFs in various aqueous energy devices, including zinc-based batteries, potassium-ion batteries, and supercapacitors [56]. Recently, Deng and coworkers reviewed the recent progress of MOF/MOF-derived nanomaterials in the cathodes, anodes, and electrolytes of AZIBs [1]. Although some previous reviews on energy-storage applications have mentioned MOFs and their derivatives [1,56], to the best of our knowledge, a critical review exclusively focusing on the application of MOF-based materials in the surface modification of the zinc metal anode has not been reported. Therefore, a critical review of the research progress on MOF-based materials for the surface modification of the zinc metal anode is essential to provide important information and valuable prospects for future research.

Herein, we review recent advances in metal–organic frameworks for the surface modification of the zinc metal anode. As shown in Figure 1, we classify them according to the element type of metal nodes because the use of different metal nodes determines the different properties of MOFs [57]. For example, Zn-based MOFs exhibit high thermal stability, large endospores, as well as a specific surface area. Cu-based MOFs have a

large specific surface area, a diverse structure, as well as unsaturated coordination metal centers. Zr-based MOFs show a better thermal, mechanical, as well as hydrolytic stability, and Ti-based MOFs have fascinating structural topologies, low toxicity, as well as high stability. In these MOF-based materials for the surface modification of the Zn anode, Zn-based MOFs and Zr-based MOFs are widely adopted, which benefit from the common features, including easy synthesis, modification, and functionalization. Furthermore, the relationships between nano/microstructures, synthetic methods of MOF-based materials, and the enhanced electrochemical performance of the zinc metal anode via MOF surface modification are systematically summarized and discussed. Finally, we propose the existing problems and possible solutions of the surface modification of the zinc anode using MOFs. We expect to deeply understand the mechanisms of various strategies to solve the problems existing in the Zn anode of AZIBs so as to provide scientific guidance for the improvement of the Zn anode performance through MOFs in the future.

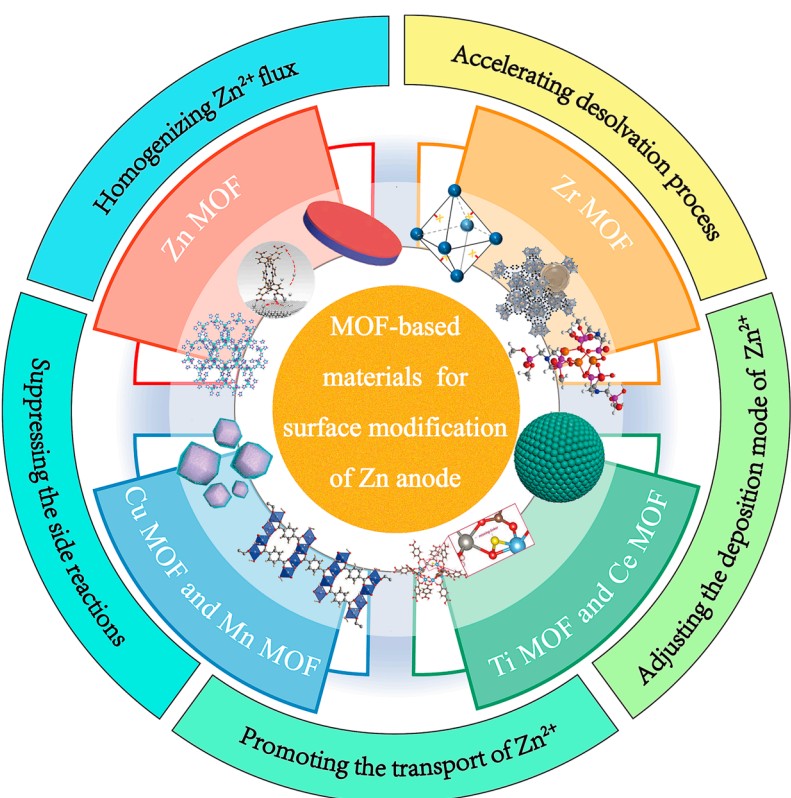

**Figure 1.** Schematic illustration of MOF-based materials for the surface modification of the Zn metal anode.

## 2. Zn-Based MOF

As a kind of MOF material, Zn MOFs have a deep research foundation in terms of structure and synthesis. In general, Zn MOFs use ZnO or ZnO octahedral clusters as the metal nodes, which can be combined with different organic ligands to obtain a different pore size, stability, or other special properties [58]. Moreover, the Zn MOF derivatives obtained by the calcination of Zn MOFs could acquire new properties, which further expand the scope of the application of Zn MOFs. In recent years, Zn MOFs and their derivatives have been widely used in the surface modification of zinc anodes [50]. The electrochemical performances of various Zn-based MOF-modified Zn anodes are shown in Table 1. In the following discussion, we categorize and summarize these studies according to the action mechanisms of Zn MOFs, including ordered porous structures and functional organic ligands.

### 2.1. Pristine Zn MOF

The high porosity, uniform pore distribution, and suitable pore size of MOFs can homogenize the $Zn^{2+}$ flux and accelerate the desolvation process of $Zn^{2+}$, thus resulting in a dendrite-free zinc anode [50]. For example, Yang et al. successfully prepared a supersaturated electrolyte layer on the surface of Zn foil by means of a ZIF-7 coating [50]. The symmetric cell with an MOF-coated Zn electrode showed an ultralong lifespan of over 3000 h at 0.5 mA cm$^{-2}$ (Figure 2c). With a 4.2 mg cm$^{-2}$ MnO$_2$ mass loading on the cathode, the MOF-coated Zn anode exhibited the specific capacity of 192.4 mAh g$^{-1}$ at the 20th cycle, and a capacity retention of 94.4% after 180 cycles at 500 mA g$^{-1}$ (Figure 2d). Due to the strong solute–solvent interactions, $Zn^{2+}$ exist in the form of complex ions in the electrolyte [59] and undergo a desolvation process before obtaining electrons, which reduces the reaction kinetics and induces side reactions. With narrower channels than solvated metal ions, the Zn MOF layer can effectively promote the solvated metal ions to remove or drop off parts of their sheath solvents, and consequently lead to desolvated electrolytes as host sieves with subnanometer-level sieving abilities (Figure 2b), which seems to be an impossible situation, even in the saturated ZnSO$_4$ solutions (Figure 2a) [60]. Lu and coworkers reported a new strategy for constructing the ion modulation layer on the Zn anode via the in situ growth of ZIF-8 [61]. The well-ordered nanochannels of ZIF-8 effectively promote the uniform distribution of the zinc ion flux, and the insulating feature of ZIF-8 is beneficial to inhibit the occurrence of side reactions. Therefore, the capacity retention of the Zn@ZIF-8||LaVO$_4$ battery can be retained up to 101%, even after 10,000 cycles, with an average CE of 99.8% at 10 mA cm$^{-2}$. Moreover, Luo et al. constructed a seamless interphase to modify the Zn anode surface by a vapor–solid reaction [62]. Through vacuum heating, vaporized 2-methylimidazole reacts with the ZnO passivation layer covered with Zn foil by deprotonation and coordinates with $Zn^{2+}$ to form ZIF-8. The MOF insulation layer hinders the vapor–solid reaction, which results in a further growth concentrated on the uncovered surface, and ultimately forming a thin (700 nm) and highly continuous layer (Figure 2e). The SEM image of MOF-Zn exhibits the tightly stacked ZIF-8 with no visible cracks (Figure 2f). The coulombic efficiency of the MOF-Cu in the Zn||Cu half-battery achieved an average value of 99.7% for 3200 cycles at 1 mA cm$^{-2}$ with a capacity of 0.5 mAh cm$^{-2}$ (Figure 2g). When assembled to the Zn||MnO$_2$ battery, the cell with the MOF-Zn anode demonstrated a capacity retention of 95% after 200 cycles at 0.5 A g$^{-1}$ (Figure 2h). This excellent performance could be ascribed to the seamless interphase preventing electrolyte diffusion in the intergranular space of the MOFs and self-corrosion. In addition, similar to the work of Yang et al., the angstrom apertures adjusted the $Zn^{2+}$ solvation structure with an oversaturated solvation structure. Similarly, Wang and coworkers fabricated a seamless interphase on the Zn anode with MOFs through the on-site chemical coordination between $Zn^{2+}$ and $[Fe(CN)_6]^{3-}$ [63]. This unique 3D open framework is beneficial for the uniform distribution of the $Zn^{2+}$ flux and rapid $Zn^{2+}$ transport kinetics. As a result, the MOF@Zn||MnO$_2$ battery exhibited an 83% capacity retention after 1500 cycles at 2 A g$^{-1}$ for long-term cycling performance. Wang et al. prepared Zn-BTC using Zn(NO$_3$)$_2\cdot$H$_2$O and 1,3,5-benzenetricarboxylate (H$_3$BTC) as the raw materials for the surface modification of the zinc anode [64]. The Zn-BTC@Zn||MnO$_2$ battery showed a high-capacity retention of 81.1% after 1000 cycles at 2 A g$^{-1}$. Moreover, He et al. used a simple strategy to synthesize ZIF-L@Zn by soaking Zn foil in a mixture of 2-dimethylimidazole and HNO$_3$ [65]. The corresponding symmetrical battery with the ZIF-L@Zn electrode could cycle for 800 h at 0.25 mA cm$^{-2}$ for 0.25 mAh cm$^{-2}$.

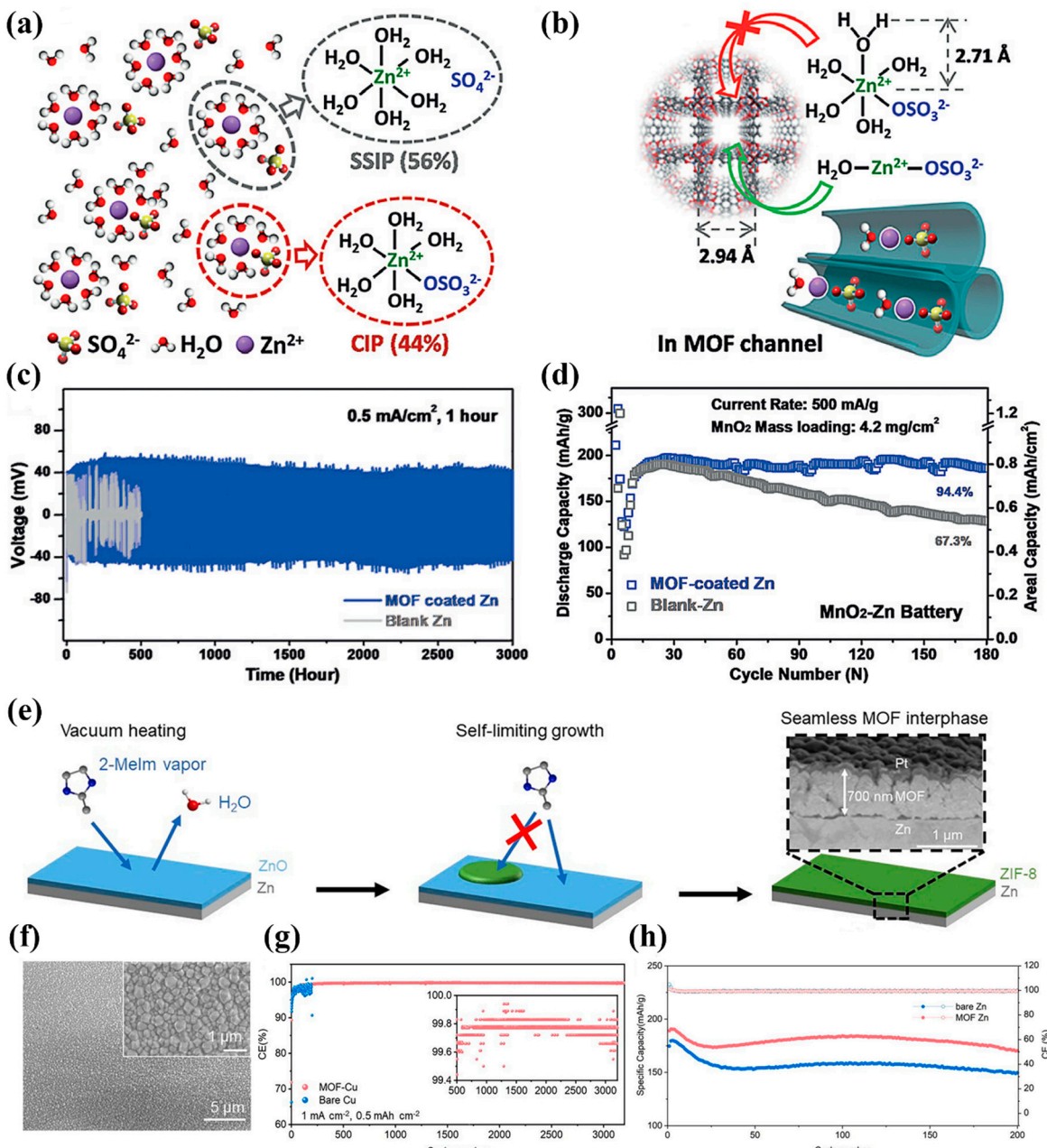

**Figure 2.** Schematic diagram of (**a**) two solvation structures in saturated (3.3 M) $ZnSO_4$ aqueous electrolytes and (**b**) highly coordinated ion complexes of $H_2O\text{-}Zn^{2+}\cdot OSO_3^{2-}$ migrating through MOF channels. (**c**) Long-term cyclic test of symmetric cells assembled by bare Zn and MOF-coated Zn anodes in 2 M $ZnSO_4$ aqueous electrolyte at 0.5 mA cm$^{-2}$ for 1 h. (**d**) Cyclic performance of $MnO_2$-Zn cells at 500 mA g$^{-1}$. (**e**) Schematic illustration of the fabrication of the seamless MOF interphase. Inset: Cross-sectional scanning electron microscopy (SEM) image of MOF-Zn. (**f**) SEM image of MOF-Zn. Inset: High magnification SEM image of MOF-Zn. (**g**) Coulombic efficiency of MOF-Cu in Zn||Cu cells at 1 mA cm$^{-2}$ with a capacity of 0.5 mAh cm$^{-2}$. (**h**) Cycling performance of Zn||MnO$_2$ cells at 0.5 A g$^{-1}$. (**a**–**d**) are adapted with permission [50]. Copyright 2020, Wiley-VCH. (**e**–**h**) are adapted with permission [62]. Copyright 2022, Elsevier B.V.

Moreover, adjusting the deposition mode of $Zn^{2+}$ by constructing special porous structures with MOFs on the surface of the Zn anode is also an effective way to obtain the dendrite-free zinc metal anode [66]. For instance, Yang and coworkers constructed 2D MOF nanoarrays on the zinc anode via a self-template method [66]. Zn foil immersed in

a tetra-(4-carboxyphenyl) porphyrin (TCPP) solution was oxidized and assembled with TCPP into Zn-TCPP on the surface of Zn (Zn-TCPP@Zn) (Figure 3a). The SEM image in Figure 3b shows that the average flake thickness of the MOF nanoarrays is about 20 nm. When tested for the full cell, the Zn-TCPP@Zn||ZVO battery exhibited a high specific capacity of 112 mAh g$^{-1}$ and 229 mAh g$^{-1}$ at the current density of 10 A g$^{-1}$ and 1 A g$^{-1}$, respectively (Figure 3d). And, at 4 A g$^{-1}$, the battery with the Zn-TCPP@Zn electrode showed a preferable capacity retention rate of 82.5% and CE of 99.9% after 1000 cycles (Figure 3e). The excellent electrochemical performance can be ascribed to the fact that, under the action of abundant zincophilic sites, such as sites containing O- and N-, Zn$^{2+}$ uniformly preseeds on the nanoarrays surface, which facilitates the preferential deposition of Zn on the Zn-TCPP nanoarrays and the subsequent lateral deposition (Figure 3c). Moreover, the insulating properties of Zn-TCPP leads to the bottom-up deposition process and ultimately to the dendrite-free Zn anode [66].

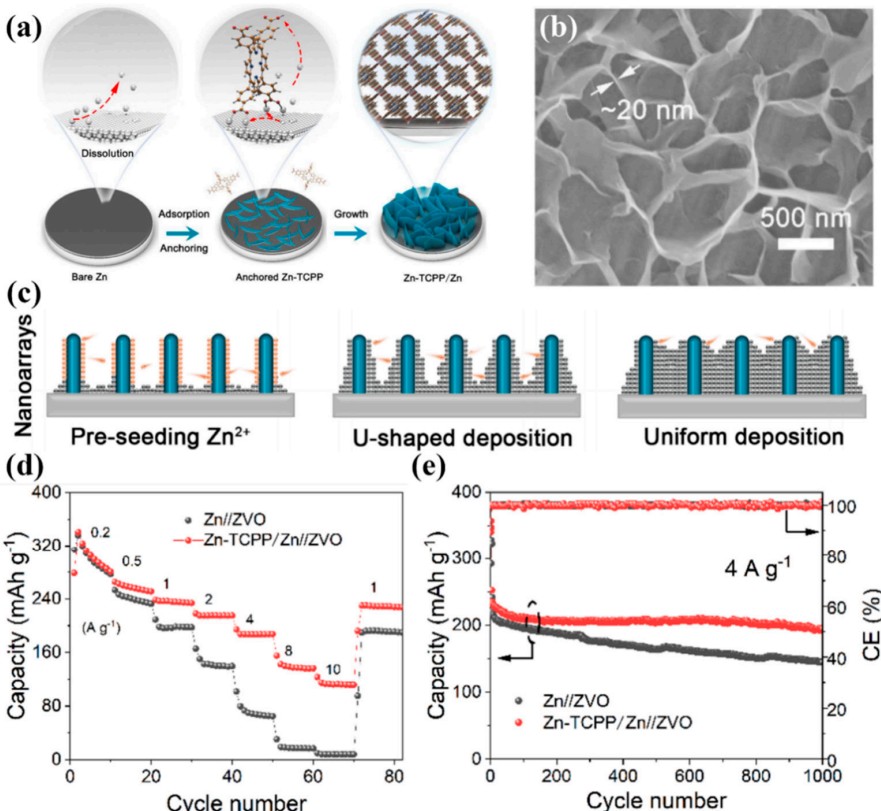

**Figure 3.** (**a**) Schematic illustration of the synthesis process of Zn-TCPP@Zn. (**b**) SEM image of Zn-TCPP@Zn. (**c**) Schematic diagram of U-shaped Zn deposition on Zn-TCPP@Zn. (**d**) Rate performance of Zn-TCPP@Zn and the Zn electrode at different current densities from 0.2 to 10 A g$^{-1}$. (**e**) Cycling performance of Zn-TCPP@Zn||ZVO and the Zn||ZVO battery at a current density of 4 A g$^{-1}$. (**a–e**) are adapted with permission [66]. Copyright 2022, Elsevier B.V.

In addition, the organic ligands in MOFs provide an alternative route for the introduction of functional groups [67]. By changing the type of organic ligands in the preparation of MOFs, different groups can be introduced into the structure of MOFs, which will expand the function of MOFs in the surface modification of the Zn anode. For example, Wang et al. designed an anionic MOF-based artificial solid-electrolyte interphase (ASEI) on the Zn anode [51]. First, Zn foil was etched by an ammonium persulfate solution in order to improve the reactivity; then, it was immersed in the precursor solution of the MOF (Zn-stp-bpy, denoted as ZSB). Through a simple wet chemical process, ZSB@Zn anode densely packed with MOF nanoparticles can be obtained (Figure 4a). As shown in Figure 4b, the diffraction peaks of the ZSB@Zn anode at 6.3° and 7.7° are assigned to the ZSB MOF

(020) and (200) diffractions, which is conducive to confirm the successful growth of the ZSB ASEI. The SEM image of ZSB@Zn shows that the surface is continuous and uniform with densely stacked MOF nanoparticles inside (Figure 4c). By virtue of the ZSB ASEI, the deposited Zn on ZSB@Zn is much denser and homogeneous. Moreover, the MOF layer was intact during the Zn deposition, as confirmed by the compact and smooth surface in Figure 4d–f. The ZSB@Zn || $Na_2V_6O_{16}$ battery could deliver a high specific capacity of 213 mAh $g^{-1}$ at 10 A $g^{-1}$ and return to the initial level when resetting to 1 A $g^{-1}$ (Figure 4g). An excellent capacity of 179 mAh $g^{-1}$ could be retained after 2000 cycles with only a 0.11‰ capacity fading rate per cycle (Figure 4h). The excellent electrochemical performance could be ascribed to two aspects: First, the use of monosodium 2-sulfoterephthalate (stp) with sulfonate groups in the precursor solution introduces sulfonate groups into the structure of ZSB, so that the pores of ZSB have dangling sulfonate groups. With a small energy barrier between the adjacent group (0.78 eV) and the lower binding energy of $Zn^{2+}$ (−1.41 eV), the dangling sulfonate groups lie in the ZBS channels and can promote $Zn^{2+}$ transferring fast and evenly, as well as accelerate the reaction kinetics on the anode; Second, ZSB combined with the Zn substrate through chemical bonds that could be Zn-N, which enables the ASEI to remain intact during cycling.

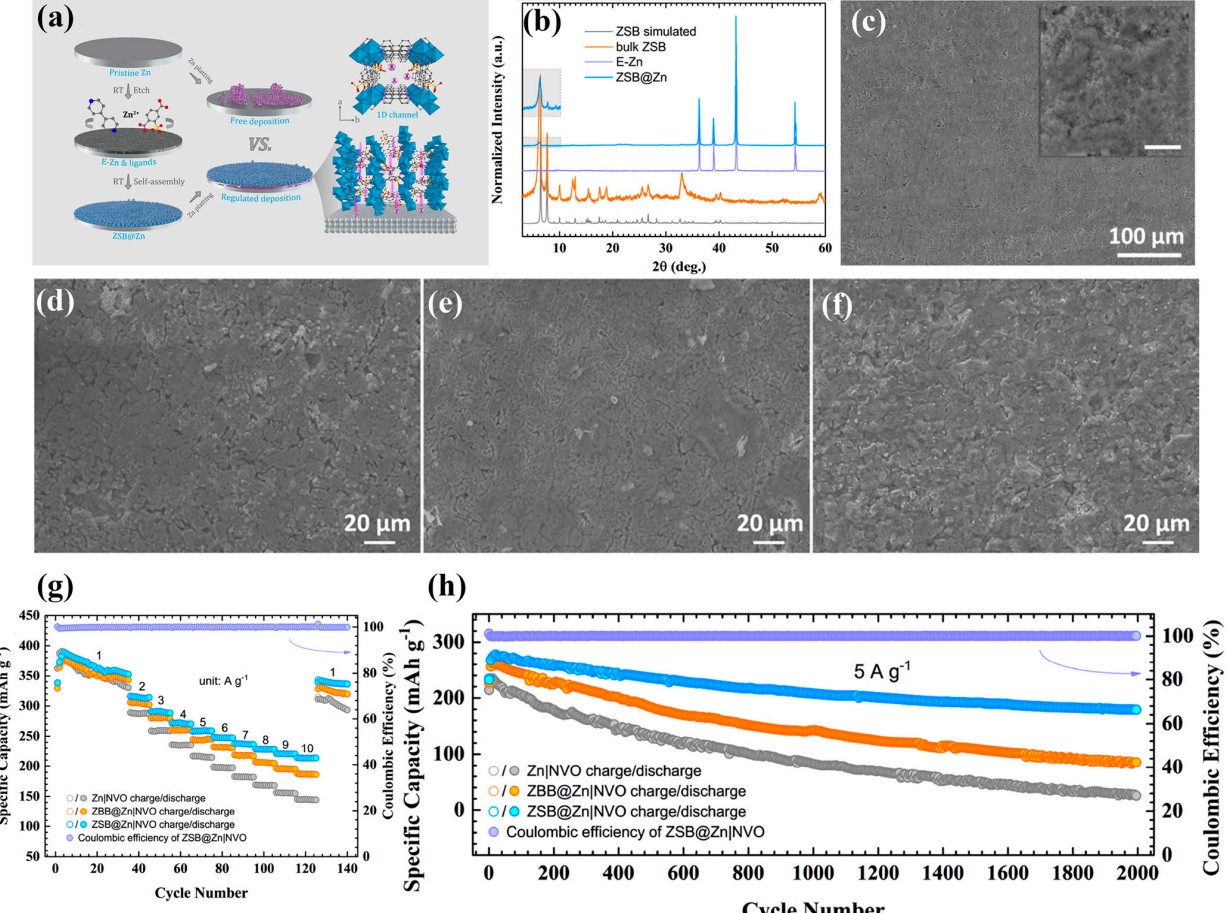

**Figure 4.** (**a**) Schematic diagram of the design and synthesis process of the ZSB@Zn anode with anionic-MOF-based artificial solid electrolyte interphase. (**b**) XRD patterns of the simulated ZSB, bulk ZSB, E-Zn, and ZSB@Zn. (**c**) SEM image of ZSB@Zn. The scale bar of the insets is 10 μm. SEM images of the ZSB@Zn anode after Zn deposition at 10 mA $cm^{-2}$ for (**d**) 5, (**e**) 10, and (**f**) 15 min, respectively. (**g**) Rate capability of the Zn, ZBB, and ZSB electrode. (**h**) Electrochemical cycling stability of the NVO full cells at a current density of 5 A $g^{-1}$. (**a**–**h**) are adapted with permission [51]. Copyright 2022, American Chemical Society.

## 2.2. Zn MOF Derivatives

In addition to pristine MOFs, MOF derivatives obtained by annealing MOFs or MOF composites are also often used for the zinc anode's performance. After pyrolysis, MOF derivatives receive a higher charge transfer ability and robustness with the preservation of ordered porous structures [68]. For example, Yuksel et al. in situ fabricated ZIF-8 on the zinc electrode by the wet chemistry method, and further pyrolyzed it to obtain N-doped porous carbon-Zn (C/Zn) [69]. The full cell with the C/Zn anode depicted a good cycling performance and a high coulombic efficiency when $MnO_2$ nanowires (NWs) were used as the cathode material, which can be attributed to the fact that the C/Zn anode not only retains the original hydrophilic and porous structure of ZIF-8, but also possesses N sites and oxygen functional groups to promote the transport of $Zn^{2+}$ [69]. Later, carbon-coated CuZn alloy nanosheets (CuZn@C NSs) were designed by in situ growing ZIF-8 on the 2D CuO substrate and with further annealing in $H_2$/Ar atmosphere [70]. When assembled with $V_2O_5$ nanobelts as the cathode material, the battery with the CuZn@C NSs anode retained a capacity of 105.4 mA h$^{-1}$ even after 1000 cycles at 5 A g$^{-1}$, which can be ascribed to the large number of zincophilic sites in the CuZn@C NSs originating from the formation of the Cu-Zn and Zn-N bond during the pyrolysis process [70].

Based on the above studies, by using the Zn MOF as the surface modification materials of the Zn anode, the electrochemical performance of AZIBs can be effectively improved. Zn MOFs can not only promote the $Zn^{2+}$ desolvation process and homogenize the $Zn^{2+}$ flux via the ordered porous structure, but also accelerate $Zn^{2+}$ transport through functional groups introduced by changing the organic ligands. Furthermore, the use of Zn MOF derivatives is able to improve the charge transfer capacity and obtain more zincophilic sites, which is conducive to the transport of electrons and ions.

## 3. Zr-Based MOF

Since being first synthesized in 2008, Zr MOFs have received increasing attention from researchers due to their unique properties [71]. $Zr^{4+}$, as a high oxidation state with four valences, has a high electron density and requires more ligand coordination [57], thus showing excellent stability. Thus, the functionality of Zr MOFs can be extended in a variety of ways, such as defect engineering. Similar to Zn MOFs, we also classify and summarize the studies on Zr MOFs according to the action mechanisms of Zr MOFs, including ordered porous structures, functional organic ligands, and MOFs as anion hosts.

For instance, Wu and coworkers synthesized a two-dimensional UiO-67 MOF (UiO-67-2D) by $Zr^{4+}$ and biphenyldicarboxylic acid for the protection of the Zn anode [72]. The UiO-67-2D@Zn||$Mn_2O_3$/C battery exhibited a high capacity of 240 mAh g$^{-1}$ at 1 A g$^{-1}$ and an 81% capacity retention after 1500 cycles at 2 A g$^{-1}$. The excellent electrochemical performance could be ascribed to the vertically arranged channels in the UiO-67-2D regulating the diffusion of $Zn^{2+}$ and accelerating the transport of $Zn^{2+}$, as well as the Zr-OH/$H_2O$ groups acting as zincophilic sites to promote uniform zinc deposition [72]. Liu et al. used nanosized UiO-66 MOFs to construct an artificial composite protective layer on the Zn anode [73]. The nanowetting effect could greatly reduce charge-transfer resistance. Thus, stable cycling for more than 500 times can be achieved by the cell with the modified Zn electrode at 3 mA cm$^{-2}$.

As one of the widely used Zr MOFs, the UIO branch is highly receptive to the introduction of functional groups [74]. Therefore, UIO-66 can obtain specific functions by introducing functional organic ligands into the structure [75]. For instance, Wang and coworkers constructed a solid electrolyte interphase with UIO-66-$SO_3$H prepared by replacing BDC (benzene-1,4-dicarboxylic acid) with BDC-$SO_3$Na during the synthesis of UIO-66 (Figure 5a) and the SPEEK binder (USL) [52]. The peaks at 1027 cm$^{-1}$ and 1170 cm$^{-1}$ in Figure 5b, and the appearance of S peaks at 167.8 eV in Figure 5c, are conducted to confirm the effective introduction of -$SO_3$H. The SEM image in Figure 5d shows that the USL coating is dense and compact. In the cyclic voltammogram (CV) curve, a stronger current response and smaller interval of redox pairs between A1 and C1 are obtained by



the USL-Zn | | VO$_2$ battery (Figure 5e). In the long-term cycling test at 1 A g$^{-1}$, the capacity of the battery with the USL-Zn electrode after five activations is 169.6 mAh g$^{-1}$ and the capacity retention reaches 95.9% after 1200 cycles (Figure 5f). This improved performance could be ascribed to the fact that the negatively charged -SO$_3^-$ groups originating from the UIO-66-SO$_3$H could effectively homogenize the zinc ion flux and adjust the desolvation process. Similarly, using the UiO-66(Zr)-(COOH)$_2$ synthesized from ZrCl$_4$ and 1,2,4,5-benzenetetracarboxylic acid (BTEC), Kim et al. fabricated a composite protective layer on the Zn anode [76]. As a result, the symmetric cell exhibited more than 2400 cycles without short-circuit at 10 mA cm$^{-2}$. Compared with the high-crystallinity MOFs reported in the literature, amorphous MOFs (aMOFs) have more active sites and a faster charge transfer due to the unsaturated coordination [77]. Furthermore, aMOFs still maintain a local structure parallel to their crystalline counterparts, while exhibiting disorder characteristics [78]. By means of the solvothermal method with Zr$^{4+}$ and nitrilotri(methylphosphonic acid), Ren et al. fabricated an amorphous MOF of ATMP-Zr (AZ) as the ASEI for the Zn anode [53]. As shown in Figure 5g, AZ particles adhere to each other, indicating the high surface energy of AZ. Brunauer–Emmett–Teller (BET) plots of two materials are displayed in Figure 5h,i. The specific areas of the AZ and AZ$_{1200}$ are 328.9 m$^2$ g$^{-1}$ and 1.7 m$^2$ g$^{-1}$, respectively. The dominant microporosity, with an average diameter of 0.53 nm (Figure 5i), may come from the intrinsic porous structure of the texture of AZ. AZs were coated on the Zn substrate with PVDF as the binder. It can be easily seen from Figure 5j and k that the AZ coating layer on the Zn is complete and dense. As a result, the AZ-Zn | | MVO battery with Mn-doped V$_2$O$_5$ (MVO) as a cathode material exhibited a specific capacity of 240 mAh g$^{-1}$ at 1A g$^{-1}$ and 130 mAh g$^{-1}$ at 10 A g$^{-1}$ (Figure 5l), respectively. As shown in Figure 5m, the AZ-Zn/MVO cell retained 130 mAh g$^{-1}$ even after 2000 cycles at 1 A g$^{-1}$. The improved performance could be attributed to two aspects: First, the micropores of the aMOF can be used as sieve pores to promote the desolvation of [Zn(H$_2$O)$_6$]$^{2+}$ and the reregulation of the Zn$^{2+}$ flux; Second, different from bare Zn, Zn$^{2+}$ migrate through the Zn$^{2+}$-hopping mechanism, attributed to dangling bonds (-PO$_3$H$^-$ or -PO$_3^{2-}$) in the aMOFs, which can remarkably enhance the transport of zinc ions. Under the synergistic effect of the above factors, the fast transference of Zn$^{2+}$ and dendrite-free Zn deposition can be realized.

In addition to ordered porous structures and functional organic ligands, Zr MOFs can also act as an anion host to encage functional anions in the electrolyte in the pores, thereby regulating the Zn$^{2+}$ flux and Zn$^{2+}$ transport process [79]. Recently, a quasisolid artificial electrolyte interphase synthesized by defective UIO-66 (D-UIO-66) for a dendrite-free Zn anode was reported [79]. The synthetic route of the D-UiO-66 layer is shown in Figure 6a. By soaking the synthesized UIO-66 in dilute hydrochloric acid for acid treatment, D-UIO-66 was obtained. Then, a slurry was made by D-UIO-66, dropped on Zn foil, and rolled repeatedly to obtain the D-UIO-66@Zn anode. In the D-UIO-66, missing linkers resulting from the acid treatment provide positively charged oxygen vacancies, which can anchor the anions present in the electrolyte by the Lewis acidic sites. In the rate-performance test, the cell with the D-UIO-66@Zn anode depicted the initial capacity of 276.8 mAh g$^{-1}$ at 0.5C (1C, corresponding to 380 mAh g$^{-1}$) and returned to 261.0 mAh g$^{-1}$ when the current rates returned to 0.5C (Figure 6b). The cell with the D-UIO-66@Zn anode also showed a high capacity of ~111.0 mAh g$^{-1}$, with about 100% CE after 2500 cycles at 5C (Figure 6c). The excellent electrochemical performance could be ascribed to the MOF channels fixed with anions, which can facilitate Zn$^{2+}$ transport and thus guide homogeneous Zn$^{2+}$ distribution. Moreover, the higher concentration of Zn$^{2+}$ in the interphase allows the consumed zinc ion on the electrolyte/electrode interface to be replenished in time, and therefore the mitigation of concentration polarization can be achieved.

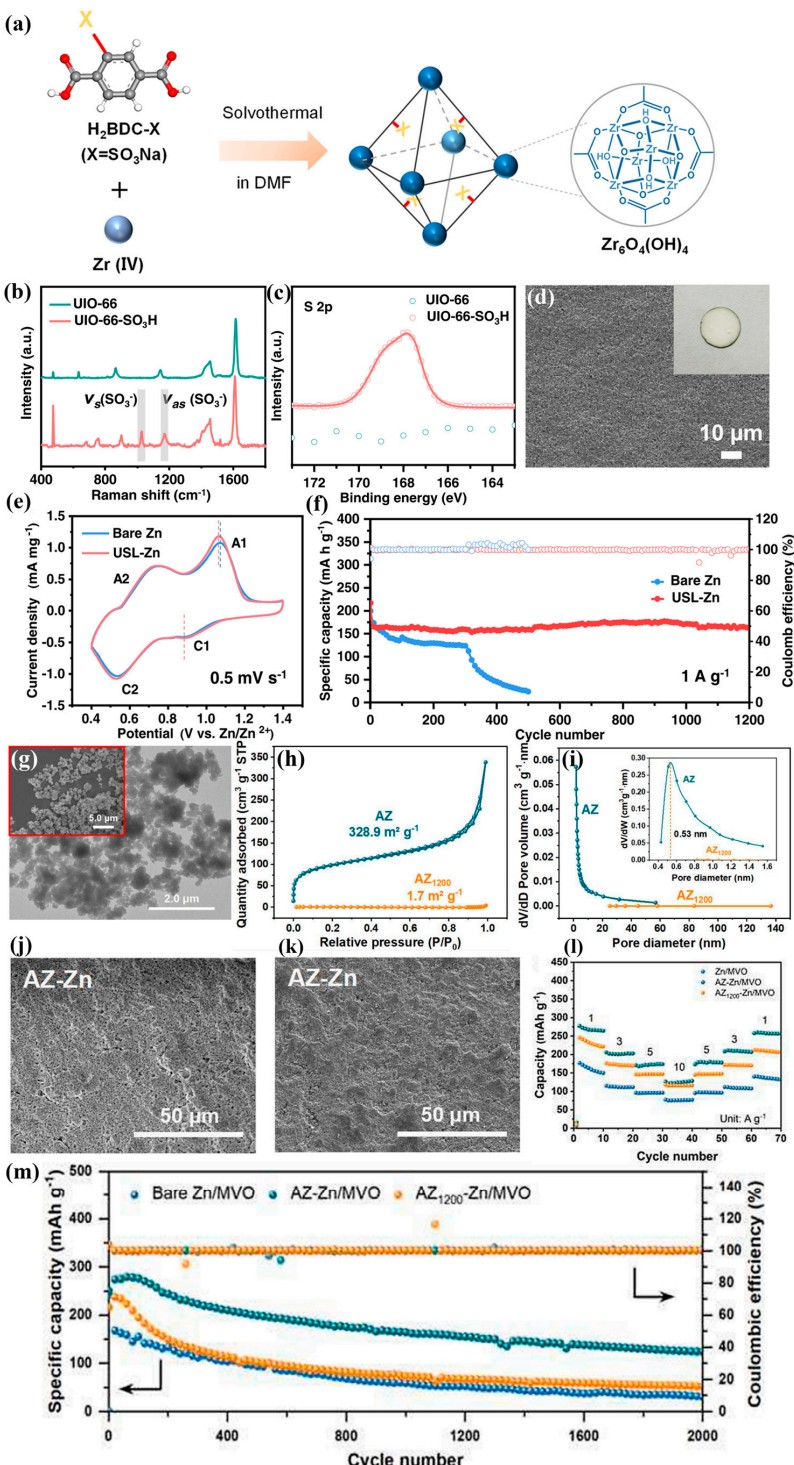

**Figure 5.** (**a**) Schematic illustration of the preparation process of UIO-66-X. (**b**) Raman spectra and (**c**) high-resolution S 2p XPS spectra of UIO-66 and UIO-66-SO$_3$H. (**d**) SEM image of USL-Zn (inset shows the optical image). (**e**) Cyclic voltammogram (CV) profiles of bare Zn and USL-Zn anode at 0.5 mV s$^{-1}$ and (**f**) Long cycle stability of Zn || VO$_2$ full cells with bare Zn and USL-Zn electrodes at 1 A g$^{-1}$. (**g**) Transmission electron microscopy (TEM) (insert: SEM) image of AZ. (**h**) BET adsorption/desorption isotherm. (**i**) Pore volume distribution. SEM images of the AZ-Zn (**j**) before and (**k**) after immersed in 2.0 M ZnSO4 after 5 days, respectively; (**l**) Rate capability of bare Zn, AZ-Zn, and AZ1200 electrodes, respectively. (**m**) Cycle stabilities of bare Zn, AZ-Zn, and AZ1200 electrodes at 1 A g$^{-1}$ (vs. MVO). (**a–f**) are adapted with permission [52]. Copyright 2022, Elsevier Ltd. (**g–m**) are adapted with permission [53]. Copyright 2023, Elsevier B.V.

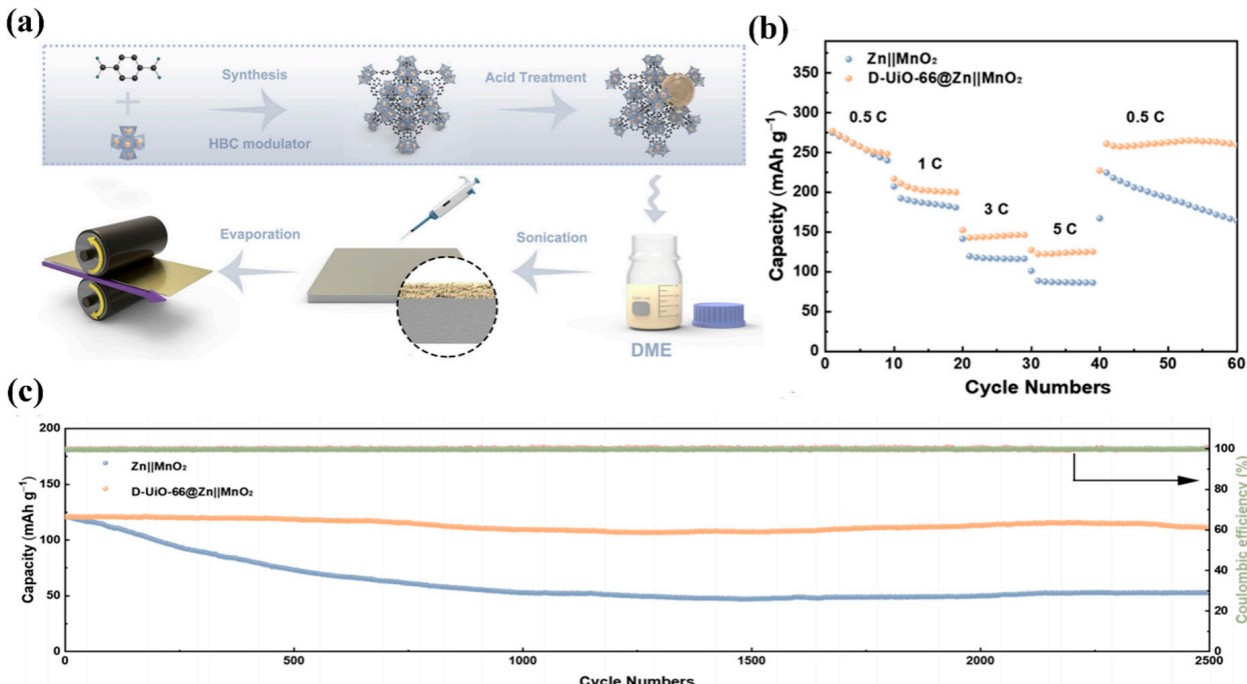

**Figure 6.** (**a**) Schematic illustration for the synthesis of the D-UIO-66@Zn anode. Zn‖MnO₂ full cells with bare Zn and D-UIO-66@Zn anode of (**b**) rate performance and (**c**) cycling stability at 5C. (**a–c**) are adapted with permission [79]. Copyright 2023, The Author(s).

Based on the above research results, the electrochemical performance of AZIBs can be improved by the Zr-MOF-modified Zn anode. The high stability of the Zr MOF allows it to withstand a wider variety of treatments. The Zr MOF can not only improve the performance of the Zn anode through the ordered porous structure and the introduction of functional groups, but also improve the ion transport behavior as anion hosts by introducing defects in the structure.

## 4. Other Typical MOF

In addition to the abovementioned Zn MOFs and Zr MOFs, other typical MOFs have been explored for the surface modification of the Zn anode, such as the Cu MOF [80,81], Mn MOF [82], Ti MOF [83,84], and Ce MOF [85]. Among them, the synthesis paths of the Cu MOF and Mn MOF are relatively cheaper and more environment friendly due to the sufficient resources and lower toxicity of Cu and Mn [86], while the unique characteristics of the Ti MOF and Ce MOF provide them with excellent development potential.

For example, Cao et al. built a hydrophobic ASEI with a self-healing function on a Zn anode via coating a $Cu_3(BTC)_2$ MOF filled with $Zn(TFSI)_2$-tris(2,2,2-trifluoroethyl)-phosphate (TFEP), which can be ascribed to the fact that the organophilic $Cu_3(BTC)_2$ MOF is organophilic and has continuous three-dimensional nanopores. Therefore, $Zn(TFSI)_2$-TFEP can easily be infiltrated and confined inside the MOF to protect the Zn anode [81]. The ASEI structure is shown in Figure 7a, and the SEM image shows a good crystal structure and a uniform Cu MOF particle size of about 0.5 μm (Figure 7b). Consequently, the Zn‖MnO₂ battery with $Zn(TFSI)_2$-TFEP protected the Zn anode and showed a high capacity of 270 mAh g$^{-1}$ at 0.5 C, and retains 135 mAh g$^{-1}$ even at the rate of 10 C (Figure 7c). At the same current density and a 2:1 capacity ratio of Zn:MnO₂, the Zn‖MnO₂ battery with Zn(TFSI)2-TFEP protected the Zn anode and retained a high discharge capacity of 141 mAh g$^{-1}$ and a capacity retention of 97.2% after 600 cycles, with a CE closing to 100% (Figure 7d). This excellent performance could be attributed to three aspects: First, benefiting from immiscibility with the aqueous electrolyte, the $Zn(TFSI)_2$-TFEP organic electrolyte confined in the MOF pores impedes water penetration and displaces solvated H₂O by the

TFEP on the electrolyte/electrode interface; Second, part of the Zn(TFSI)$_2$-TFEP is reduced to form a ZnF$_2$-Zn$_3$(PO$_4$)$_2$ ASEI under the Zn plating potential, which further decreases the direct water contact with the Zn anode; Third, the broken ZnF$_2$-Zn$_3$(PO$_4$)$_2$ ASEI can be repaired by reducing Zn(TFSI)$_2$-TFEP, and thus the failure of the ASEI resulting from the breakage can be avoided. Postmodification of the synthesized Cu MOF is also an effective strategy. For instance, Sun and coworkers in situ grew a Cu-MOF on zinc foil and successfully prepared selenized Cu-MOF@Zn (SCM@Zn) by a thermal selenization process [80]. Benefitting from the improvement of zincophilicity ascribed to selenization, the SCM@Zn||ZVO battery retained a high capacity of 198.8 mAh g$^{-1}$ after 800 cycles at 5.0 A g$^{-1}$. Recently, Cao et al. improved the performance of the Zn anode by constructing an angstrom ionic sieve membrane based on the 2D Mn-MOF [82]. With CaV$_8$O$_{20}$·nH$_2$O as the cathode, the battery with the 2D Mn-MOF@Zn electrode showed the capacity of ~150 mAh g$^{-1}$ during initial cycles and an increasing trend during subsequent cycles, with a 119.5% capacity retention after 1000 cycles at 4 A g$^{-1}$.

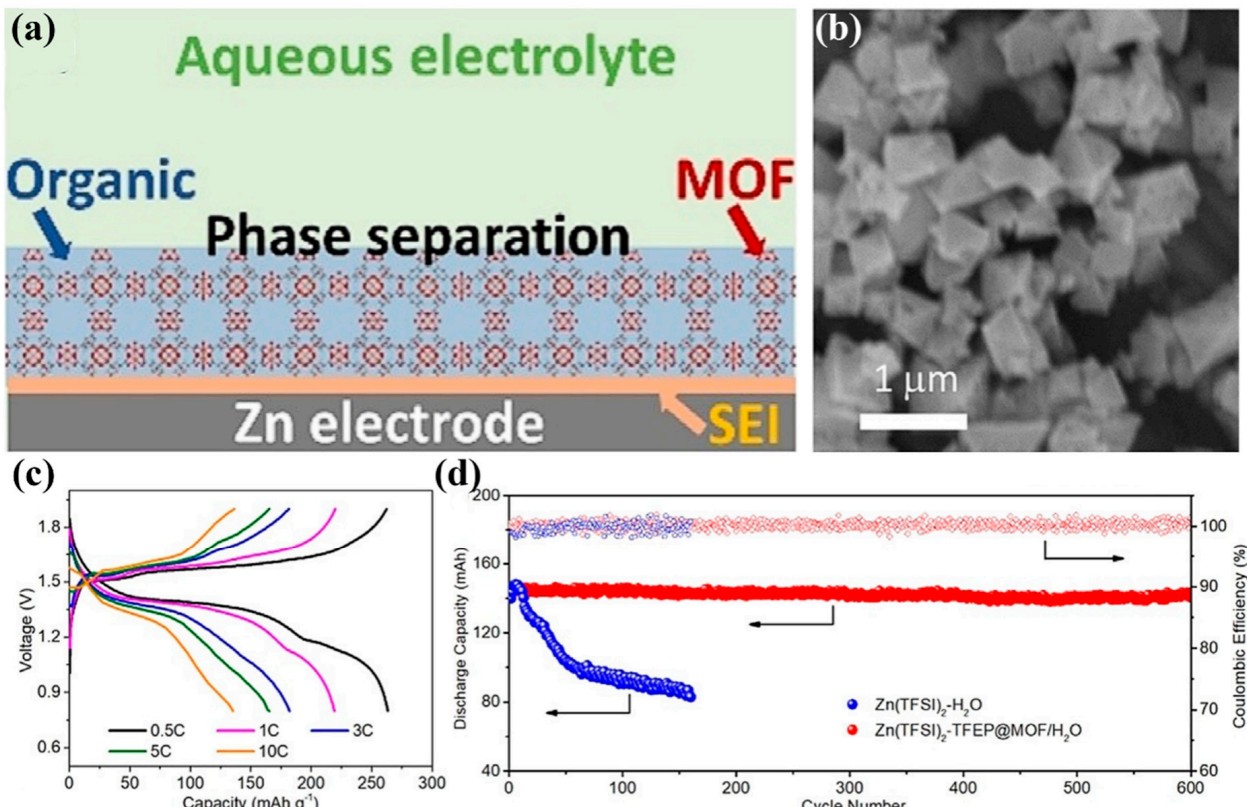

**Figure 7.** (**a**) Schematic illustration of solid electrolyte interphase formation in the MOF confined organic electrolyte. (**b**) SEM image of MOF. (**c**) Rate performance of Zn(TFSI)$_2$-TFEP protected Zn anode. (**d**) Long cycling performance and CE curves of the Zn||MnO$_2$ cells with bare Zn and Zn(TFSI)$_2$-TFEP protected Zn anode at 10 C. (**a**–**d**) are adapted with permission [81]. Copyright 2020, Wiley-VCH.

In recent years, Ti MOFs have attracted wide attention because of their good electrochemical properties. As an element belonging to the same main group as Zr, the MOFs composed of a Ti node also exhibit high stability. Meanwhile, Ti MOFs have the advantages of a high natural abundance and low toxicity and redox properties when compared with Zr MOFs [87]. By in situ doping MIL-125(Ti) with a small amount of Zn, Zhao et al. fabricated bifunctional MIL-125(Ti)-Zn with missing linkers to realize the inhibition of dendrites and the hydrogen evolution reaction [84]. The Zn deposition behavior of the MIL-125(Ti)-Zn@Zn anode is shown in Figure 8a. The morphology of MIL-125(Ti)-Zn has the same morphology as MIL-125(Ti) (Figure 8b). Zn and Ti are obviously present in MIL125(Ti)-Zn

(Figure 8c–f). The MIL-125(Ti)-Zn@Zn symmetrical cell showed a voltage hysteresis as low as 80 mV for 2100 h at the current density of 1 mA cm$^{-2}$ (Figure 8g). Furthermore, the MIL-125(Ti)-Zn@Zn-Cu half-cell exhibited an average CE value of 99.01% after 600 cycles at 1 mA cm$^{-2}$ (Figure 8h). The excellent performance could be attributed to two aspects: First, produced by the partial substitution of Ti with Zn, the missing linkers could guide the transfer of electrons from Ti to O, which allows the formation of electron-rich oxygen sites. The presence of electron-rich O can greatly reduce the diffusion barrier of Zn$^{2+}$ and therefore accelerate the diffusion kinetics and uniformize Zn$^{2+}$ flux, finally achieving homogeneous Zn deposition; Second, with a strong adsorption on H*, electron-rich oxygen sites are beneficial to restrain the Heyrovsky reaction or Tafel reaction, both of which are an important part of the hydrogen evolution reaction, so the HER can be effectively suppressed [88,89]. Lately, inspired by the easy hydrolysis properties of MIL-125 (Ti), Zou and coworkers in situ constructed an in situ growth high dense TiO$_{2-x}$ solid electrolyte interphase (HDSEI) on the zinc anode through a process in which the MIL-125 (Ti) coating layer was hydrolyzed during the cycles [83]. A symmetric battery with the HDSEI@Zn electrode showed a cycling performance of over 4200 h and 1300 h under 1 mA cm$^{-2}$ and 3 mA cm$^{-2}$, respectively.

Similarly, the Ce MOFs with Ce$^{4+}$ as the metal node also have good stability. Furthermore, CeO$_2$ has a corrosion-protective ability [90] as well as a high affinity for Zn$^{2+}$ [91], which are two crucial factors for the high performance Zn anode coating. For instance, Li and coworkers used Ce-MOF-808 as a precursor to fabricate defect-rich MOF-CeO$_2$ by pyrolysis at 500 °C, and coated it on the Zn anode by a spin-coating process (Figure 8i) [85]. MOF-CeO$_2$ inherits the morphology of the Ce-MOF-808 with a mean size of 150 nm (Figure 8j). With manganese-expanded hydrated vanadate (MnVO) as the cathode, the MOF-CeO$_2$@Zn||MnVO battery exhibited high overlap charge and discharge curves, indicating good reversibility and stability (Figure 8l). In the cycling performance test at 5 A g$^{-1}$, the MOF-CeO$_2$@Zn||MnVO battery maintained a capacity of 163 mAh g$^{-1}$ after 10,000 cycles, with a CE closing to 100% (Figure 8m). The reasons for the excellent electrochemical performance are explained in Figure 8k. For the MOF-CeO$_2$, the affinity to Zn and lattice defects introduced by pyrolysis act as active sites to accelerate the transport of Zn$^{2+}$. The negatively charged characteristics of the coating layer shields anions and attracts cations, which is conducive to suppressing side reactions. Moreover, the pore structure of the MOF-CeO$_2$ facilitates the desolvation process of Zn[(H$_2$O)$_6$]$^{2+}$ and reregulates the Zn$^{2+}$ flux. The above research results show that the capacity and cycle life of AZIBs can also be effectively improved by using other types of MOFs, which is performed with no treatment or treated by filling pores with hydrophobic electrolytes, through selenization, the introduction of defects, hydrolysis, or pyrolysis as a protective layer for Zn anode.

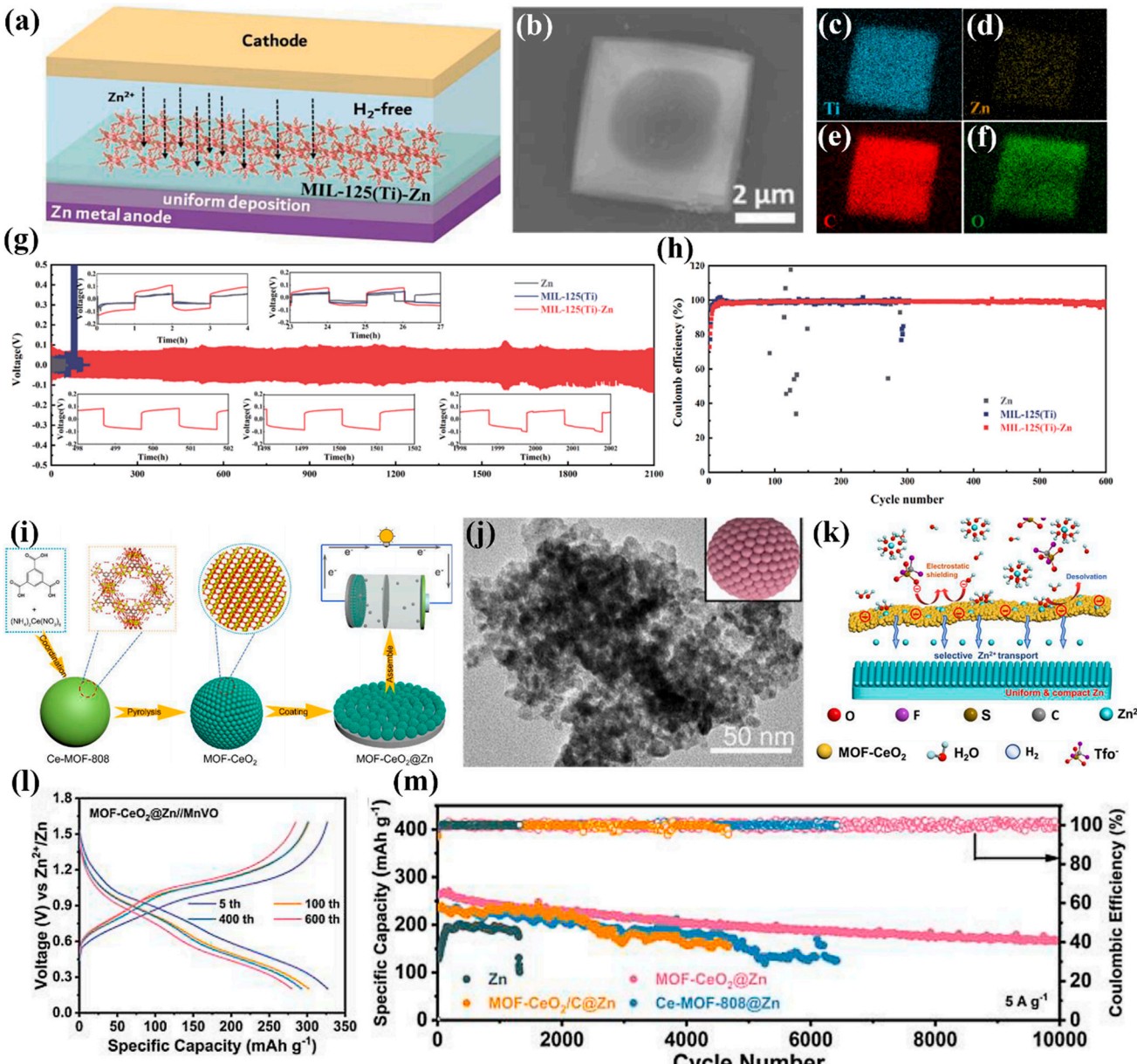

**Figure 8.** (**a**) Schematic illustration of zinc-deposition process on the MIL-125 (Ti)-Zn@Zn metal anode. (**b**) SEM and (**c–f**) energy-dispersive spectroscopic (EDS) elemental mapping images of MIL-125(Ti)-Zn powder, Ti element (blue), Zn element (yellow), C element (red), and O element (green). (**g**) Voltage profiles of symmetric cells assembled by bare Zn foil and MIL-125(Ti)-Zn@Zn anodes at 1mA cm$^{-2}$ and 1 mAh cm$^{-2}$. (**h**) CE of the Zn plating/stripping on bare Zn, MIL-125(Ti)-Zn, and MIL-125(Ti)-Zn@Zn anodes at 1mA cm$^{-2}$ and 1 mAh cm$^{-2}$. (**i**) Schematic diagram of the preparation of MOF-CeO$_2$ and MOF-CeO$_2$@Zn. (**j**) TEM image of MOF-CeO$_2$. (**k**) Schematic diagram of the product of Zn deposition at MOF-CeO$_2$@Zn anode. (**l**) Charge–discharge profiles of the MOF-CeO$_2$@Zn anode. (**m**) Long cycle stability of MOF-CeO$_2$@Zn anode at 5 A g$^{-1}$. (**a–h**) are adapted with permission [84]. Copyright 2022, Elsevier B.V. (**i–m**) are adapted with permission [85]. Copyright 2022, Elsevier B.V.

**Table 1.** Electrochemical performance of various types of MOFs as surface modification materials for the Zn anode.

| Interfacial Layers | Preparation Methods | Voltage Hysteresis $V$ [a] (V) ($C_1$ [c] (mA cm$^{-2}$)) | Lifespan $T$ [b] h [$C_1$ [c] (mA cm$^{-2}$), $C_2$ [c] (mAh cm$^{-2}$)] | Ref. |
|---|---|---|---|---|
| | | Zn MOF | | |
| ZIF-7 | Doctor blading method | 0.028 (0.3) | 3000 (0.5,0.5) | [50] |
| ZIF-8 | One-step solvent thermal method | 0.058 (2) | 1200 (2,1) | [61] |
| ZIF-8 | Vapor–solid reaction | / | 3000 (1,1) | [62] |
| Zn-MOF | On-site chemical coordination | 0.080 (4) | 2100 (4,2) | [63] |
| Zn-BTC | Doctor blade method | / | 400 (4,1) | [64] |
| Zn-TCPP | Self-template method | 0.050 (5) | 2600 (0.2,0.2) | [66] |
| ZSB | In situ growth | / | 3100 (5,2.5) | [51] |
| ZIF-L | In situ growth | 0.026 (0.25) | 800 (0.25,0.25) | [65] |
| ZIF-8 derived carbon materials | In situ growth and pyrolyzing | / | / | [69] |
| CuZn@C | Coating | ~0.025 (2) | 500 (1,1) | [70] |
| | | Zr MOF | | |
| UIO-67-2D | Drop-casting | 0.018 (0.25) | 850 (0.5,0.5) | [72] |
| UIO-66-SO$_3$H | Drop-casting | ~0.040 (1) | 700 (5,5) | [52] |
| UIO-66-(COOH)$_2$ | Manually coating | 0.080 (10) | 240 (10,/) | [76] |
| ATMP-Zr | Doctor blade method | 0.032 (1) | 1800 (1,1) | [53] |
| D-UIO-66 | Rolling process | 0.066 (1) | 1800 (1,1) | [79] |
| UIO-66 | Coating | / | 100 (3,3) | [73] |
| | | Other Typical MOF | | |
| Cu$_3$(BTC)$_2$ MOF filled with Zn(TFSI)$_2$-TFEP@Zn | Coating | 0.020 (0.5) | 700 (0.5,0.5) | [81] |
| SCM | Hydrothermal and selenization | / | 500 (2,1) | [80] |
| 2D Mn-MOF | Squeegee method | ~0.0446 mV (4) | 2000 (4,4) | [82] |
| MIL-125(Ti)-Zn | Coating | ~0.080 (1) | 2100 (1,1) | [84] |
| MIL-125(Ti) | Scraper method | / | 4200 (1,1) | [83] |
| MOF-CeO$_2$ | Spin-coating | / | 3200 (3,1) | [85] |

[a] Voltage (V). [b] Time (h). [c] $C_1$: current density (mA cm$^{-2}$); $C_2$: specific area capacity (mAh cm$^{-2}$).

## 5. Conclusions and Outlook

In conclusion, we have summarized the research progress on the applications of MOF-based materials for the surface modification of the Zn anode. The electrochemical performance of various MOF-modified Zn anodes are presented in Table 1. Several kinds of MOFs are outlined, such as the Zn MOF, Zr MOF, Cu MOF, Ti MOF, and so on. Among them, the Zn MOF and Zr MOF are highlighted. As a kind of MOF-based material that has been studied for a long time, the Zn MOF has been comprehensively understood and investigated by researchers, and there are many ways to treat it, such as using different ligands to construct Zn MOFs, so as to obtain MOFs with the required properties. Compared to the Zn MOF, which uses $Zn^{2+}$ as the metal node, the recently developed Zr MOF can coordinate more ligands due to the higher valence state of $Zr^{4+}$, so it has better stability, which can make the Zr MOFs withstand defect engineering. These MOF materials show great potential as the interface modification layer of the Zn anode, which include: (1) a large number of active sites to accelerate the transport of $Zn^{2+}$; (2) suitable pore structures, which are conducive to the desolvation process of $Zn[(H_2O)_6]^{2+}$; (3) the adjustment of the $Zn^{2+}$ flux to induce uniform zinc deposition. Furthermore, by introducing functional groups or defects, MOFs could obtain new properties. In addition, some unique methods, such as constructing two-dimensional nanoarrays or ASEIs with a self-healing function using MOFs, have also achieved good performance. These results show that, as a sort of coating

material for the Zn anode, MOF-based materials can effectively improve the cycle stability and cycle life of AZIBs.

Up to now, many research studies have been carried out for preparing ASEIs on the Zn anode surface with MOFs, which could be divided into two categories, including ex situ coating techniques and in situ growth techniques. For the ex situ coating technique, there are three main methods, including the doctor blading method, spin-coating method, and drop-casting method, which all have the characteristics of being simple processes and having a low cost. Among them, the doctor blading method is the most commonly used method to construct coatings on Zn anodes with a well-defined thickness (mainly from 10 to 500 μm). It has the advantage of good repeatability and to a certain extent hinders mass transfer during cycling. By contrast, the spin-coating method, by which uniform ASEIs could be constructed by centrifugal force, can be used to build thinner coatings (<20 μm) and decrease the restriction of the $Zn^{2+}$ transport. However, this method is difficult to apply to large-scale fabrication because the raw material utilization rate is low (2%~5%), as well as due to the limitation of the substrate area. Moreover, the drop-casting method is to drop liquid droplets on the Zn substrate and spread them to form a film with a thickness on a micro- to nanometer scale, and its raw material utilization rate is close to 100%. But, the nonuniformity of the coating and the poor controllability of the thickness make this method limited for large-scale application. Although the above ex situ methods can form ASEIs with a low cost, simple process, and controllable composition as well as thickness on the Zn anode, these methods still have some shortcomings, such as the use of toxic solvents during the preparation of part of the coating not being closely combined with the Zn substrate, which can be well solved by the in situ growth method, thus avoiding the failure of the coating resulting from volume changes during cycling.

Although MOFs and their derivatives have shown promising outcomes in enhancing the Zn anode performance, there are still some challenges needing to be overcome in practical applications. Here, we put forward our own perspectives on these problems.

1. Develop novel types and functions of MOFs for the surface modification of the Zn anode. Up to now, tens of thousands of MOFs have been designed and fabricated, but only a few dozen are used for the coating materials of Zn anodes in AZIBs. The research on MOFs on the surface modification of the Zn anode is still in the early stage, and more kinds of MOFs need to be explored. As a coating material used in AZIBs, MOFs should be continuously immersed in an aqueous electrolyte, which requires MOFs to have a high water stability. However, most MOFs are less stable in water systems [56]. The MOFs with a high valence metal as the node have stronger coordination bonds and a larger coordination number, which endows it a more stable structure, and the stable structure allows it to accommodate a variety of treatments. Therefore, MOFs with a hypervalent metal center may be the future development trend of Zn anode coating materials. Additionally, it is necessary to explore the new functions and properties of the MOFs that have been applied in practice.

2. Explore new composites of MOF coating materials. Although the use of bare MOFs as an interfacial protective coating material for the Zn anode has achieved good results, the exploration of MOF composite materials should be paid attention to. By combining MOFs with other functional materials, the materials with desired properties can be obtained. For example, most MOFs have a low electrical conductivity. By creating composites with conductive carbon materials, coating materials with a high conductivity can be obtained, which helps to reduce the charge accumulation and homogenize the electric field distribution on the Zn anode surface.

3. Develop reliable, convenient, and low-cost synthesis strategies. At present, most of the MOF derivatives used for the surface modification of the Zn anode are prepared by calcination, in which the MOF derivatives are prone to structural collapse [55]. Therefore, reliable preparation techniques are needed to maintain the porous structure after annealing. Additionally, cost is an important consideration for commercialization. Although the cost of zinc is low, the synthesis and modification of MOFs will increase

the material consumption and process complexity. Some organic ligands are expensive and have a low practical application value. Thus, in the preparation of MOFs, priority should be given to low-cost, environmentally friendly organic ligands and simple synthesis methods.

4.  Reveal the mechanism. AZIBs exhibit excellent performance by the surface modification of the Zn anode with MOFs, but the mechanisms of MOFs are still unclear. Most of the current characterization techniques are ex situ characterizations, which cannot well reveal the changes in the composition and structure of MOFs during the cycle. Hence, it is necessary to develop in situ characterization techniques as well as use theoretical simulation and calculation to provide an in-depth understanding and theoretical basis for the mechanism of MOFs.

In general, the application of MOFs in the surface modification of the Zn anode has been increasing in recent years, which indicates its great application potential. The method of the surface-modified Zn anode with MOFs provides a reliable solution for the sustainable development of the future of society and contributes to the drive towards "carbon neutrality". However, research on MOFs for the surface modification of the Zn anode is still in the early stages, and it has the above challenges to be solved. With the unremitting efforts of researchers, a deeper understanding of it can be acquired, which will contribute to the better design and optimization of MOF coatings. We believe that the surface modification for the zinc anode with MOFs will promote the development of high-performance AZIBs in the future.

**Author Contributions:** Conceptualization, Y.L.; methodology, Y.X., K.F. and C.K.; validation, Y.X., G.W. and Q.H.; investigation, Y.X. and Y.P.; writing—original draft preparation, Y.L. and Y.X.; writing—review and editing, Y.L. and Y.X.; funding acquisition, Y.L. All authors have read and agreed to the published version of the manuscript.

**Funding:** This work was financially supported by the National Key Research and Development Program of China (2020YFB1713500), the Open Fund of State Key Laboratory of Advanced Refractories (no. SKLAR202210), the Student Research Training Plan of Henan University of Science and Technology (2022040, 2022044, 2023040), and the Undergraduate Innovation and Entrepreneurship Training Program of Henan Province (S202110464005).

**Institutional Review Board Statement:** Not applicable.

**Informed Consent Statement:** Not applicable.

**Data Availability Statement:** Not applicable.

**Conflicts of Interest:** The authors declare no conflict of interest.

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
