# Peer review of "Recent Advances in Metal–Organic Frameworks for the Surface Modification of the Zinc Metal Anode: A Review"

_coatings, doi:10.3390/coatings13081457_

Round 1

Reviewer 1 Report

The authors provided a review of recent work on zinc metal anode modified MOFs. In general the parameters studied in this review are interesting but slight corrections are necessary before accepting this paper.

Comments

- Please present some analysis methods used for this kind of modified materials (like XRD, XPS,...).

- It is very interesting to present in the manuscript some structural and textural properties of modified MOFs.

- Please present the formation mechanism of modified MOFs.

- Table 1 is not cited in the manuscript, please cite it.

- Please discuss the results mentioned in table 1 in terms of performance and stability of materials.

Reviewer 2 Report

The review-article concentrates on application of aqueous ZnII-ion batteries as promising energy storage devices. Particularly, the readers' attentionis focused on ZnII-ion based MOFs of solvate ZnII-complexes looking at solvent water (H2O) as a ligand, in addition to surface modification reactions of Zn-anode via metal-anode processes of ligand exchange with inorganic (SO42-) and organic ligands.

Generally, the contribution would be of interest in the readers of Coatings, because of development of ZnII-solvate based batteries show low cost; they are environmentally friendly; and high safty devices, respectively.

The review-article also deals with Zr-based MOFs in addition to other type of MOFs such as Ce-, Ti- and Cu-containing materials.

The major drawback of the review-article is that, there is a little attention concentrated on heterogeneous solvent-solid surface reactions, despite that there are highlighted few examples of ligand exchange of SO42- anions with solvent H2O ligand of ZnII(H2O)62+ complex (Figure 2.) In fact, the solution chemistry of aqua-complexes of Zn2+ ion is very complex. The authors should concentrate on works [x1,x2] in the latter context.

Moreover, the ligand exchange processes are associated with change of geometry of ZnII(O)6 chromophore, but the phenomenon is not discussed by the authors.

In addition, the octahedral (Oh) geometry of the metal chromophore (page 3, row 106) does not appear typical geometry of ZnII-ion. Conversely, this is an atypical geometry, which can be found in solid-state MOFs, due to an affect of organic ligands, rather than in solution, where mainly ZnII(H2O)62+ complex exhibits Oh-geometry of its chromophore (ZnO16). The ligand-exchange reactions cause for a distortion of the Oh-geometry. For instance, chromophores of type ZnIIO14O22, ZnIIO15O2, ZnIIO13O23 do not show Oh-geometry, when O1 = H2O, while O2 = OH-, SO42-, HCOO-, et cetera, ligands. Typically, ZnII-ion shows coordination number 4 and perturbed Td-geometry of ZnIIX4-chromophore (X-inorganics, respectively, organics). Detail can be found on reference [x3].    

[x1] Ivanova, B., Spiteller, M. Electrospray ionization mass spectrometric solvate cluster and multiply charged ions: a stochastic dynamic approach to 3D structural analysis. SN Appl. Sci. 2, 731 (2020). https://doi.org/10.1007/s42452-020-2555-0.

There is not needed extensive correction of the English.

Author Response

  1. The review-article concentrates on application of aqueous ZnII-ion batteries as promising energy storage devices. Particularly, the readers' attentionis focused on ZnII-ion based MOFs of solvate ZnII-complexes looking at solvent water (H2O) as a ligand, in addition to surface modification reactions of Zn-anode via metal-anode processes of ligand exchange with inorganic (SO42-) and organic ligands.

Generally, the contribution would be of interest in the readers of Coatings, because of development of ZnII-solvate based batteries show low cost; they are environmentally friendly; and high safty devices, respectively.

The review-article also deals with Zr-based MOFs in addition to other type of MOFs such as Ce-, Ti- and Cu-containing materials.

Response: We thank the reviewer for his/her positive assessment for our work and we have carefully revised our manuscript according to the valuable comments and suggestions.

  1. The major drawback of the review-article is that, there is a little attention concentrated on heterogeneous solvent-solid surface reactions, despite that there are highlighted few examples of ligand exchange of SO42-anions with solvent H2O ligand of ZnII(H2O)62+ complex (Figure 2.) In fact, the solution chemistry of aqua-complexes of Zn2+ ion is very complex. The authors should concentrate on works [x1,x2] in the latter context.

Response: We are thankful to the reviewer for his/her comments. Following the reviewer’s advice, we have described the heterogeneous solvent-solid surface reactions in more detail in the first work. As for the following content, considering that they are the same principle as the first work, so we only briefly describe them, reading as follows:

Due to the strong solute–solvent interactions, Zn2+ exist in the form of complex ions in the electrolyte [56] and undergo a desolvation process before obtaining electrons, which reduces the reaction kinetics and induces side reactions. With narrower channels than solvated metal ions, Zn MOF layer can effectively promote the solvated metal ions to remove or drop off parts of their sheath solvents and consequently lead to desolvated electrolytes as host sieves with sub-nanometer level sieving abilities (Figure 2b), which seems like an impossible situation even in the saturated ZnSO4 solutions (Figure 2a).” Please see Page 4.

“In addition, similar to the work of Yang et al.,” Please see Page 4.

  1. Moreover, the ligand exchange processes are associated with change of geometry of ZnII(O)6chromophore, but the phenomenon is not discussed by the authors.

Response: We thank the reviewer for the valuable suggestions and comments. We tried to discuss the phenomenon of ligand exchange processes, but after carefully checking the original literature, we found that the authors did not discuss the phenomenon, and it is difficult for us to discuss them through its original description, for which we apologize.

  1. In addition, the octahedral (Oh) geometry of the metal chromophore (page 3, row 106) does not appear typical geometry of ZnII-ion. Conversely, this is an atypical geometry, which can be found in solid-state MOFs, due to an affect of organic ligands, rather than in solution, where mainly ZnII(H2O)62+complex exhibits Oh-geometry of its chromophore (ZnO16). The ligand-exchange reactions cause for a distortion of the Oh-geometry. For instance, chromophores of type ZnIIO14O22, ZnIIO15O2, ZnIIO13O23 do not show Oh-geometry, when O1 = H2O, while O2 = OH-, SO42-, HCOO-et cetera, ligands. Typically, ZnII-ion shows coordination number 4 and perturbed Td-geometry of ZnIIX4-chromophore (X-inorganics, respectively, organics). Detail can be found on reference [x3].

Response: We are thankful for the reviewer’s valuable suggestions and comments. The octahedral (Oh) geometry of the metal chromophore (page 3, row 106) does not refer to solvated Zn2+, but clusters acting as the metal nodes of MOFs. We apologize that we are not sure about what the reviewer’s instruction is.

Due to the lack of annotation in reference x2 and x3, the modifications we made may not fully conform to the reviewer’s opinions, for which we apologize.

Reviewer 3 Report

Xing and coworkers summarized recent advances of modification of  Zn anode by utilizing metal-organic frameworks in aqueous zinc ion batteries. Multiple MOFs based on different metals, the fabrication methods, as well s MOF derivatives are well discussed. After addressing the following minor issues, I would recommend for a publication. 

1) Before classifying different metals, the authors should justify the differences in different metal-based MOFs. Why Zn/Zr-based MOFs are widely adopted and other metal-based MOFs are less used.

2) The authors listed preparation methods in Table 1. Some words are needed to discuss the advantages or disadvantages of each preparation methods. 

Author Response

Reviewer 3#: Xing and coworkers summarized recent advances of modification of Zn anode by utilizing metal-organic frameworks in aqueous zinc ion batteries. Multiple MOFs based on different metals, the fabrication methods, as well s MOF derivatives are well discussed. After addressing the following minor issues, I would recommend for a publication.

Response: We thank the reviewer for his/her positive assessment, in particularin particular, and his/her recommending our article for publication. We have carefully revised our manuscript according to the valuable comments and suggestions.

1) Before classifying different metals, the authors should justify the differences in different metal-based MOFs. Why Zn/Zr-based MOFs are widely adopted and other metal-based MOFs are less used.

Response: We thank the reviewer for his/her valuable comments and constructive suggestion. We have added a brief introduction to the differences among the different metal-based MOFs and explained why Zn/Zr-based MOFs are widely adopted when compared with other metal-based MOFs, reading as follows.

For example, Zn-based MOFs exhibit high thermal stability, large endospores as well as specific surface area. Cu-based MOFs have a large specific surface area, diverse structure as well as unsaturated coordination metal centers. Zr-based MOFs show a better thermal, mechanical as well as hydrolytic stability and Ti-based MOFs have fascinating structural topologies low toxicity as well as high stability. In these MOF-based materials for the surface modification of Zn anode, Zn-based MOFs and Zr-based MOFs are widely adopted, which is benefited from the common features including easy synthesis, modi-fication and functionalization. Please see Page 2 and 3.

2) The authors listed preparation methods in Table 1. Some words are needed to discuss the advantages or disadvantages of each preparation methods.

Response: We acknowledge the reviewer for his/her constructive suggestions.

Following the reviewer’s advice, we have compared the advantages and disadvantages of different preparation methods for coatings in the revised manuscript, reading as follows:

Up to now, many research studies have been carried out for preparing ASEI on Zn anode surface with MOFs, which could be divided into two categories, including ex-situ coating techniques and in-situ growth techniques. For ex-situ coating technique, there are three main methods, including doctor blading method, spin-coating method and drop-casting method, which all have the characteristics of simple process and low cost. Among them, the doctor blading method is the most common used method to construct coatings on Zn anode with a well-defined thickness (mainly from 10 to 500 μm). It has the advantages of good repeatability and certain extent hinders mass transfer during cycling. By contrast, the spin-coating method, by which uniform ASEIs could be constructed by centrifugal force, can be used to build thinner coatings (<20 μm) and decrease the re-striction of Zn2+ transport. However, this method is difficult to apply to the large-scale fabrication, because the raw material utilization rate is low (2~5%) and the limitation of the substrate area. Moreover, the drop-casting method is to drop liquid droplets on the Zn substrate and spread to form a film with thickness on micro- to nanometer scale, and its raw material utilization rate is close to 100%. But the nonuniformity of the coating and the poor controllability of the thickness make this method limited for large-scale application. Although the above ex-situ methods can form ASEI with low cost, simple process, and controllable composition as well as thickness on Zn anode, these methods still have some shortcomings such as the use of toxic solvents during the preparation of part of the coating, not closely combined with the Zn substrate, which can be well solved by in situ growth method, thus avoiding the failure of the coating resulting from volume changes during cycling.” Please see Page 17.
